# Preliminary Approach for the Development of Sustainable University Campuses: A Case Study Based on the Mitigation of Greenhouse Gas Emissions

Ivo Araújo, Leonel J. R. Nunes and António Curado *

proMetheus, Instituto Politécnico de Viana do Castelo, Rua da Escola Industrial e Comercial de Nun'Alvares, 4900-347 Viana do Castelo, Portugal
* Correspondence: acurado@estg.ipvc.pt

**Abstract:** University campuses consume a significant amount of energy. Given the high volume of people who commute to and from campuses, the resources consumed, such as water and energy, and the amount of waste that must be managed, they can be compared to small towns. To address this issue, university managers and decision-makers have implemented various technical measures to improve water and energy efficiency and waste management. These measures aim to increase campus sustainability and enhance the well-being of the academic community. One popular measure is the installation of autonomous energy production systems, such as photovoltaic (PV) systems, which replace external energy sources and reduce GHG emissions. For example, a PV system installed on a university campus has been found to supply 19% of the campus's electricity needs and replace 21 $tCO_2 \cdot yr^{-1}$. However, adopting organizational measures to manage the use of produced energy and increasing school community's environmental awareness about energy efficiency is crucial in order to change behavior and improve campus sustainability.

**Keywords:** sustainable university campus; PV energy; renewable energy sources; carbon footprint





## 1. Introduction

University campuses are built infrastructures set up in a given territory, with a decisive role in developing the communities where they are implanted, contributing and, therefore, establishing clusters that promote the spreading of knowledge [1–4]. If, on the one hand, university campuses can contribute significantly to the development of communities through this function of creating and disseminating knowledge [5], then, on the other hand, due to the enormous scale that campuses can reach, they can become centers of high-levels of energy use, high-levels of waste production, and high-levels of water consumption, as well as high-levels of maintenance requirements for buildings, infrastructures, and green areas [6–8]. Additionally, the need to reinforce the transportation capacity to provide a large number of students with a safe and reliable service to travel around on and off university campuses can cause serious management issues, especially if the objective is to meet the needs of sustainable development, given the high rate of private motor vehicle use which persists strongly in many countries [9–12]. The use of cleaner transportation options is mandatory, from biking to walking, as well as the planning of an efficient public transportation network designed to serve the local university community [13,14]. As mentioned by Alshuwaikhat et al. [15], university campuses can be considered small cities due to their size, their population, and the complexity of the activities that take place within, which can cause several direct and indirect impacts on the environment [16]. Lozano et al. [17] mentioned that universities have historically played important roles in transforming societies and training decision-makers, leaders, entrepreneurs, and academics. However, it seems that some difficulties are found in the self-application of sustainability principles

to campus management [18]. Despite all of the knowledge developed in universities concerning sustainability, circular economy, or energy efficiency, implementing management practices and the organization of campuses have encountered constraints that hinder their sustainable development [19–23].

However, universities are the place to find solutions to problems, and they will undoubtedly be the right places to debate, study, and find solutions in the form of management models or simply individual actions to contribute to the sustainability of their campuses [24,25]. In the same way, Amaral et al. [26] note that, given the need to address the challenges of climate change actively, university leaders and managers are showing a growing interest in reducing the environmental impacts of their campuses. As the authors emphasize, this approach can be categorized into energy, buildings, water, transportation, land use and occupation, air quality, climate, and food, on which key actions to reduce negative impacts should be prioritized. For example, increasing energy production on campuses and decreasing energy consumption in buildings are the leading measures to be implemented [27]. However, as the authors also mention, with little dissemination of the positive impact caused by these measures, the image of university campuses as large-scale inefficient energy consumers can hardly be improved. Another study, presented by Ferreira et al. [28], addresses the research problem of determining if relevant energy and water savings may be obtained on university campuses without significant investments based on technical and organizational measures. In other words, the authors argue that it is often possible to significantly change habits to save energy and water consumption without having to undertake costly infrastructure investments.

As previously mentioned, the installation of renewable energy self-production systems on university campuses seems to be one of the most recommended solutions, and is found in several previous studies [29–33]. Among the options, installing photovoltaic (PV) systems seems to attract many decision-makers, given the growing number of initiatives recently described. For example, Lee et al. [34] showed deliverables on the economic feasibility of campus-wide PV systems in New England (USA), demonstrating that the economic parameters validated the investment worthiness. Alternatively, Akwa et al. [35], Khan et al. [36], Herrando et al. [37], Alghamdi et al. [38], Grisales-Noreña et al. [39], Enano et al. [40], and Tarigan [41] focused more specifically on aspects related to production and consumption, as well as the reduction of the carbon footprint by replacing fossil energy sources by renewable energy sources.

The reduction of carbon emissions is also one of the main concerns of university campus managers [42,43]. Lim and Hayder [44] mentioned that the emission of greenhouse gases (GHG) is an alarming issue in the world today, causing not only the rise in temperature but also the occurrence of natural disasters such as floods, hurricanes, droughts, and others. This concern with the reduction of emissions has led to the application of measures distributed across four primary areas of intervention: water and waste management, transportation, energy, and buildings. In addition to organizational and technical interventions, the same authors note that higher education institutions must address the diverse needs of local societies and promote sustainability. In other words, universities will be able to serve as an example by presenting management models and functional technical solutions on their campuses, which other sectors of society can replicate [45–49]. Zhen et al. [50] go a little further, suggesting that universities are responsible for leading ecological civilization and low carbon transition. The authors conclude that it is imperative for universities to formulate effective low-carbon policies in order to achieve sustainable development and confront global climate change.

The purpose of this research is to investigate the impact of photovoltaic (PV) systems on energy consumption and greenhouse gas emission reduction on a university campus. The study faces the challenge of limited available space for PV installation due to the heritage status of many campus buildings and restrictions on alterations. Therefore, it is essential to examine alternative organizational measures to manage energy consumption and raise environmental awareness among the university community regarding energy

efficiency and sustainable development. To gather information on the current level of community engagement with energy conservation and efficiency, a survey was conducted among the academic community using the studied campus. The survey aimed to gather qualitative and quantitative data on the community's commitment to reducing electricity consumption and energy efficiency, with a focus on reducing carbon emissions.

## 2. A Brief Literature Review

The involvement of user willingness is a crucial factor in ensuring the sustainability of university campuses [51,52]. To ensure the success of energy saving and efficiency initiatives, it is imperative to actively involve the university community in the implementation of these measures [53,54]. Surveys are an effective methodology for evaluating a community's knowledge and attitudes towards current and future measures [55,56]. In addition to conducting surveys, establishing educational and training programs that focus on sustainability and energy efficiency is crucial for the long-term success of these initiatives [57,58]. These programs aim to raise awareness and educate the university community on the significance of reducing energy consumption and increasing energy efficiency [59]. The programs may also involve the establishment of working groups and volunteer opportunities, such as implementing recycling and composting programs, which allow community members to actively contribute to the pursuit of sustainable solutions [60]. Holding events and lectures on sustainability issues can also serve as an effective method to raise community awareness and encourage discussions on important topics [61,62].

The survey results can be utilized to enhance sustainability and energy efficiency initiatives on campus and to evaluate the impact of these initiatives [63–66]. Furthermore, the survey can serve as a useful tool for identifying areas where further work is required to achieve sustainability goals [67,68]. The analysis of the survey results should include an evaluation of the community's level of engagement and the representativeness of the results which were obtained [69]. Additionally, the community's understanding of sustainability issues and their perspectives on the implemented or planned measures must also be considered [70]. The analysis should encompass the identification of trends and patterns in the results, including the community's preferences and priorities with respect to sustainability and energy efficiency [71,72]. It is imperative to determine whether the implemented measures are consistent with the community's expectations and requirements, and to identify areas where further work is necessary to satisfy the community's needs [73,74]. The assessment of user willingness through a survey is an effective methodology for ensuring the sustainability of university campuses and actively involving the university community in energy saving and efficiency initiatives [75,76].

## 3. Materials and Methods

### 3.1. Location of the Case Study Area

The Agrarian School of Ponte de Lima (ESA IPVC) campus is part of the Polytechnic Institute of Viana do Castelo, located in the parish of Refóios do Lima in the municipality of Ponte de Lima, Alto Minho region, Northern Portugal, as shown in Figure 1.

The campus comprises several buildings, with the main building being the Academic Building, where the institution's teaching activities, scientific research, and administrative services take place. The university residence, which includes support services such as a canteen, cafeteria, and laundry, is also a prominent feature of the campus. The main building is a former 17th-century monastery, originally built in the 12th century and classified as a national monument, with several restrictions on interventions. The ESA IPVC campus covers an area of approximately 17 hectares, including the experimental farm where crops such as vines, olive groves, and other crops are produced, and areas dedicated to the production of indigenous cattle breeds. Due to restrictions on interventions in the main building and its surroundings, the rooftop of the student's residence was used to install the photovoltaic (PV) system, which is formed by 119 JASOLAR Half-Cell 455W modules with a total peak power of 54,145 kWp and a 50 kW core inverter.

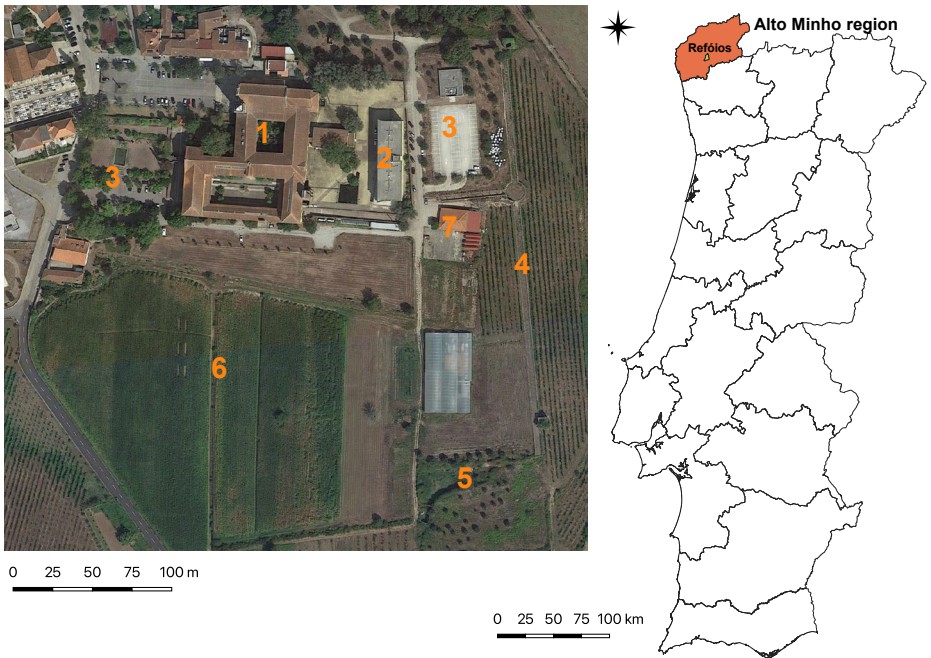

**Figure 1.** Location of the ESA IPVC campus. 1—Academic Building; 2—Students Residence; 3—Parking and Gardens; 4—Experimental Vineyard; 5—Experimental Orchards; 6—Cultivation Fields; and 7—Agricultural Production Support Buildings.

### 3.2. Power Consumption of ESA IPVC Campus

To determine the power consumption of the ESA IPVC campus, a historical record was analyzed from December 2010 to January 2020. Based on this record, the minimum and maximum estimated values for power consumption were determined, taking into consideration the variability in consumption over the past decade.

### 3.3. Calculation of PV Production System Installed on the ESA IPVC Campus

To determine the monthly potential for PV production by the system installed at the ESA IPVC campus, the PVGIS platform (available at https://joint-research-centre.ec.europa.eu, accessed on 10 May 2022) was utilized. A comparison was made between the recorded production for the months when the system was in operation and the estimated production for the months without recorded data.

### 3.4. Calculation of the Carbon Footprint for the ESA IPVC Campus

The calculation of the carbon footprint for the consumption of the ESA IPVC campus was performed using the data provided by the external energy supplier on the invoice.

### 3.5. Assessment of the Current State of Awareness and Commitment of the ESA IPVC Community with Issues Related to Energy Savings and Efficiency

The assessment of the level of commitment and awareness of the ESA IPVC community regarding energy saving and energy efficiency was carried out through a survey. The survey population consisted of students (550), teaching and research staff (40), and administrative, technical, and support staff (18). Visitors were not included in the survey. The survey was administered between 11 and 16 November 2022, and was distributed to all members of the community using the internal mailing list of ESA IPVC. A total of 101 meaningful responses were received, with a margin of error of 9%. The survey consisted of 17 questions and its aim was to gather information on the community's understanding of the measures being implemented to reduce energy consumption and improve energy efficiency, as well as their opinions on these measures and other potential measures that could be implemented in the future (Table 1).

**Table 1.** Organization of the questions presented in the survey.

| Section 1 |
| --- |
| Tell which ESA IPVC community group you belong to:<br>    Students<br>    Teaching and research staff<br>    Administrative, technical and support staff |

| Section 2 |
| --- |
| 1. Are you aware of the efforts undertaken by the IPVC to improve energy efficiency in thedifferent schools?<br>    Yes<br>    No |
| 2. Do you know that photovoltaic solar panels were installed on the roof of the ESA IPVC residence?<br>    Yes<br>    No |
| 3. Reducing energy consumption in ESA's daily activities also depends on changing individual habits. Can you give an example of small practices that can contribute to the reduction of electrical consumption at ESA IPVC? |
| 4. Within your possibilities, do you have an idea of the importance of your role in reducing ESA IPVC's electrical consumption? In other words, do you consciously think that you are already doing everything in your power to save energy?<br>    Yes<br>    No |
| 5. Do you think that an effort is being made to reduce electricity consumption on the ESA IPVC campus?<br>    Yes<br>    No |
| 6. Can you give an example of measures that are being implemented at ESA IPVC, to reduce electricity consumption? |
| 7. Do you feel that the set of measures to reduce electricity consumption that is being implemented at ESA IPVC is communicated in the best way?<br>    Yes<br>    No |
| 8. Do you think it is important that information about energy efficiency measures being implemented at ESA IPVC be communicated to the entire community?<br>    Yes<br>    No |
| 9. On a scale of 1 to 5 (where 1—Not at all Important and 5—Very Important), how do you rate the need to implement measures to improve energy efficiency at ESA IPVC? |
| 10. On a scale of 1 to 5 (where 1—Not at all Important and 5—Very Important), how would you rate the quality of the measures that are being implemented to improve energy efficiency at ESA IPVC? |
| 11. Can you provide an example of an additional measure that you would like to see implemented at ESA IPVC to improve energy efficiency on campus? |
| 12. Are you willing to play an active role in changing behavioral habits to reduce energy consumption at ESA IPVC?<br>    Yes<br>    No |
| 13. Would you be willing to give up of some comfort in order to achieve more ambitious carbon footprint reduction goals?<br>    Yes<br>    No |
| 14. Can you give an example of one of these personal sacrifices for the environment? |
| 15. Would you be available to integrate an "Awareness Brigade" for issues related to the reduction of energy consumption and energy efficiency?<br>    Yes<br>    No |
| 16. From the following measures, select the 3 that you most agree with to be additionally implemented at ESA IPVC:<br>    16.1. Conducting awareness-raising actions for all members of the ESA IPVCcommunity.<br>    16.2. Installation of motion sensors in passage areas and bathrooms, to turn on the lights only when necessary.<br>    16.3. Limit the use of elevators only to people with reduced mobility or in situations where volumes are being transported.<br>    16.4. Limit the use of non-essential electrical equipment on the ESA IPVC campus, such as radios, televisions, heaters, and fan heaters, among others, in the residence rooms, faculty cabinets, and other similar places.<br>    16.5. Adjust school schedules to make the most of the hours of natural light and to avoid the use of artificial light (whenever the situation allows). |

The first question in the survey was designed to categorize the responses based on the different groups that comprise the community, as each group has unique characteristics and differing perspectives on the issues at hand.

## 4. Results

### 4.1. Estimation Power Consumption at ESA IPVC

As defined in Section 2, the estimation of electricity consumption at ESA IPVC was carried out supported by the historical record collected between December 2010 and January 2022, as shown in Figure 2. Then, the mean and standard deviation of the minimum and maximum values were calculated, obtaining the representative values of 23,467.96 kWh·month$^{-1}$ and 30,098.74 kWh·month$^{-1}$, respectively.

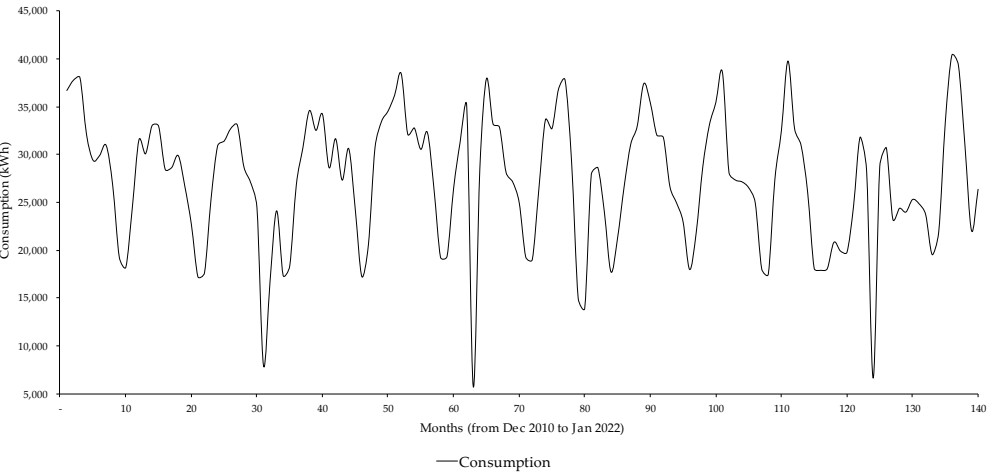

**Figure 2.** Distribution of monthly consumption data from December 2010 to January 2022. Estimation of the energy production potential for the ESA IPVC photovoltaic plant.

As previously described in Section 2, the potential for electricity production was estimated using the PVGIS platform (available at https://joint-research-centre.ec.europa. eu, accessed on 10 May 2022). This platform calculates the estimated electrical energy production of a photovoltaic plant based on technical parameters such as the location, installed power, type of panels, inclination of the panels, and estimated system losses. The results obtained for the photovoltaic plant at ESA IPVC are presented in Table 2.

**Table 2.** Estimation of the electricity production of the ESA IPVC photovoltaic plant based on the PVGIS platform.

| Month | Production (kWh) | Standard Deviation |
|-------|------------------|--------------------|
| Jan | 3247.1 | 806 |
| Feb | 4196.5 | 853.1 |
| Mar | 5773.2 | 1063.2 |
| Apr | 6573.8 | 781.5 |
| May | 7921.3 | 955.9 |
| Jun | 8101.7 | 697.3 |
| Jul | 9025.5 | 614.8 |
| Aug | 8428.7 | 617.9 |
| Sep | 6945.8 | 567.9 |
| Oct | 4873.7 | 833.7 |
| Nov | 3404.3 | 749.4 |
| Dec | 3158.5 | 469.8 |

The results obtained from the PV system's first five months of operation were analyzed to adjust the model and reduce the associated error. This was done by comparing the

estimated values from the PVGIS platform with the actual production data. The results, presented in Table 3, were obtained by assuming an average adjustment factor of 87.5% between the estimated and real values.

**Table 3.** Estimation of the real production values of the PV system.

| Month | Average | Standard Deviation | Model Adjustment | Observation |
|---|---|---|---|---|
| Jan | 3900.3 | 49.2 | 120.1% | Observed Value |
| Feb | 3772.5 | 61.8 | 89.9% | Observed Value |
| Mar | 3642.2 | 71.1 | 63.1% | Observed Value |
| Apr | 5680.3 | 58.6 | 86.4% | Observed Value |
| May | 6962.8 | | 87.9% | Estimated Value |
| Jun | 7121.4 | | 87.9% | Estimated Value |
| Jul | 7933.4 | | 87.9% | Estimated Value |
| Aug | 7408.8 | | 87.9% | Estimated Value |
| Sep | 6105.4 | | 87.9% | Estimated Value |
| Oct | 4284.0 | | 87.9% | Estimated Value |
| Nov | 2992.4 | | 87.9% | Estimated Value |
| Dec | 2530.0 | 48.0 | 80.1% | Observed Value |

As can be seen from the results, the total annual production estimated by the PVGIS platform is 71,659 kWh·yr$^{-1}$, while the actual production seems to tend towards a value close to 62,500 kWh·yr$^{-1}$. This difference corresponds to a 13% discrepancy between the value estimated by the PVGIS platform and the observed value. However, analyzing the variances and the averages of the two data sets, using the F-Snedecor and the t-Student tests, the values obtained are, respectively, 0.66 and 0.18. In the case of the F-Snedecor test, the value obtained does not reject the null hypothesis; therefore, it can be concluded that there is no difference between the supposedly equal variances. In the case of the t-Student test, the value obtained also does not reject the null hypothesis; therefore, it can be concluded that there are no significant differences between the means of the two groups. This situation can also be confirmed by the projection of the two groups of data, where the real data approach the lower limit of the standard deviation of the estimated data, represented by the bars, as can be seen in Figure 3.

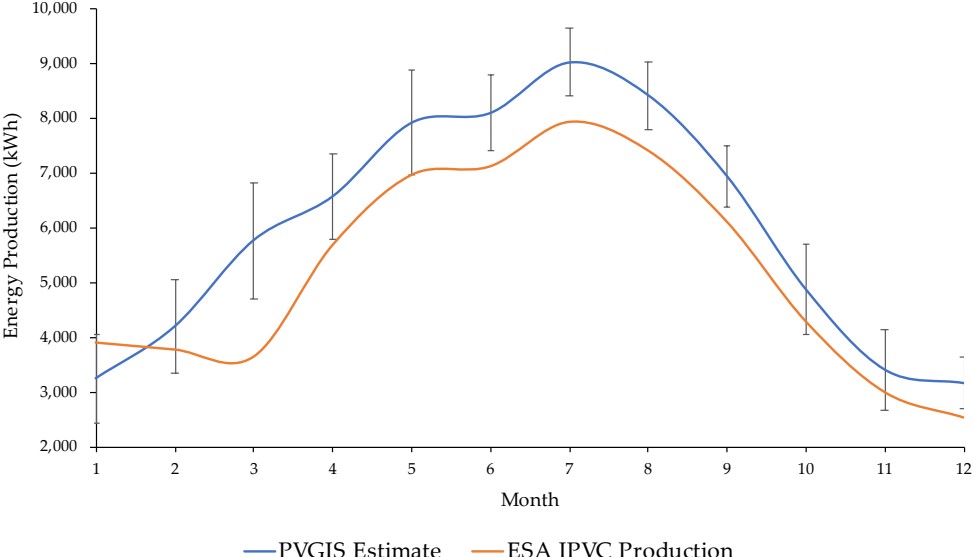

**Figure 3.** Projection of the estimated results for the electricity production and the observed results.

## 4.2. Calculation of the ESA IPVC Carbon Footprint

Based on the results presented in Sections 3.1 and 3.2, it was possible to estimate the annual consumption at ESA IPVC, which is approximately 295,000 kWh·yr$^{-1}$. Thus, with an electricity production of 62,500 kWh·yr$^{-1}$, consumption is expected to be replaced by a self-production of 21%. Based on the energy supplier's data on the origin of the electricity used at ESA IPVC, which are presented in Table 4, it is possible to calculate the carbon footprint of this energy based on the value of 480.24 gCO$_2$·kWh$^{-1}$.

**Table 4.** Energy sources and carbon footprint calculation.

| Energy Sources | Energy Sources (%) | Consumption (kWh) | CO$_2$ Emissions (gCO$_2$·kWh$^{-1}$) |
|---|---|---|---|
| Coal | 35.54% | 111,716.24 | 53,650,605 |
| Fossil CHP | 8.09% | 25,414.55 | 12,205,081 |
| Renewable CHP | 3.97% | 12,482.76 | 5,994,722 |
| Natural Gas | 21.81% | 68,550.42 | 32,920,656 |
| Eolian | 6.93% | 21,792.45 | |
| Other Renewable | 1.45% | 4550.07 | 2,185,126 |
| Hydropower | 16.76% | 52,685.04 | |
| Nuclear | 5.16% | 16,224.60 | |
| MSW | 0.29% | 898.04 | 431,275 |

The results indicate an annual emission of approximately 100 tCO$_2$·yr$^{-1}$. Using the ESA IPVC PV system, it is estimated that 21 tCO$_2$·yr$^{-1}$ will be avoided.

## 4.3. Analysis of the Results Obtained in the Survey

The answers obtained in the survey are distributed among the different groups of the ESA IPVC community as shown in Figure 4.

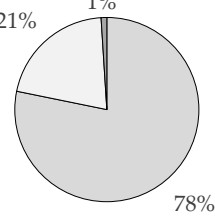

□ Students

□ Teaching and research staff

■ Administrative, technical and support staff

**Figure 4.** Distribution of survey responses by the different groups of the ESA IPVC community.

The results of the survey indicate that the student group was the largest contributor to the total responses, accounting for 78% of the valid responses. Meanwhile, the teaching and research staff group accounted for 21% of the valid responses, and the administrative, technical, and support staff groups contributed 1% of the valid responses. Despite the large number of responses from the student group, it is worth noting that the teaching and research staff group had a higher percentage of participation, with over 50% of its members responding to the survey questions.

The first two questions in the survey were designed to gauge the general knowledge of the community about the institution's efforts to improve energy efficiency in all schools under IPVC and about the recent investment in installing a PV system on the roof of the university residence. The results of these questions are illustrated in Figure 5.

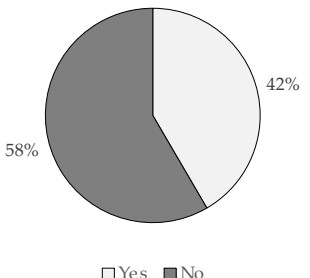
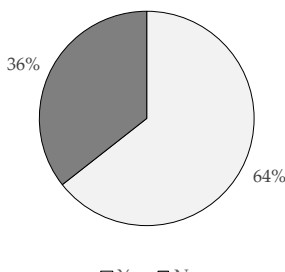

**Figure 5.** Data obtained in questions "1. Are you aware of the efforts undertaken by the IPVC to improve energy efficiency in the different schools?" and "2. Do you know that photovoltaic solar panels were installed on the roof of the ESA IPVC residence?".

As indicated by the results, a significant portion of the ESA IPVC community was not aware of the efforts made by the IPVC to improve energy efficiency in other schools. However, the community had a better understanding of the investments made in their own campus. This suggests that there is a need for greater dissemination and communication of information regarding the energy efficiency initiatives taken by the IPVC across all schools.

The third question posed in the survey, asking for examples of small practices that could contribute to reducing electrical consumption at ESA IPVC, received different types of answers. In this set of responses, the high number of respondents who were unable to present an example of these small practices stands out, representing around 15% of the total responses obtained. On the other hand, the most frequently mentioned example was "turning off the lighting in the classrooms when they are finished and in the cabinets/rooms when leaving". This opinion corresponded to approximately 70% of the total responses obtained. Other examples such as "using LED lamps", "turning off the lights in the bathrooms", "not using the elevators", or "not using auxiliary heating systems in cabinets or bedrooms" were also presented by the respondents, corresponding to about 15% of the total responses obtained.

Regarding the fourth question, in which respondents answer about their contribution to energy savings on the ESA IPVC campus, a significant part, as can be seen in Figure 6, considered themselves as having a role in this energy saving.

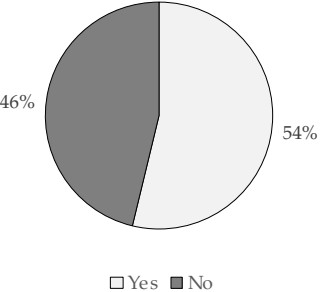

**Figure 6.** Result of the question "4. Within your possibilities, do you have any idea of the importance of your role in reducing ESA IPVC's electrical consumption? In other words, consciously, do you think you already do everything to save energy?".

From the individual analysis of the responses obtained, there seems to be a clear division between the types of responses given by different groups in the academic community. The groups of teaching and research staff and administrative, technical, and support staff present a dominance of "no" answers, indicating that they are aware that they are still not doing everything they can to save energy, while the group of students predominantly answered "yes", most likely because their ability to intervene and decide on matters is lower.

Once again, the lack of knowledge about the efforts being implemented to reduce energy consumption is evident from the answers obtained in question "5. Do you think

that an effort is being made to reduce electricity consumption on the ESA IPVC campus?" As can be seen in Figure 7, a significant percentage of community members were unaware that measures are being taken in this direction.

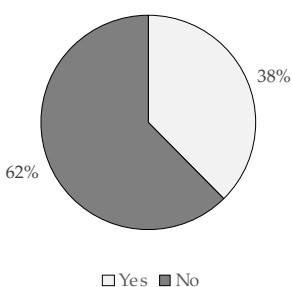

**Figure 7.** Results for question "5. Do you think that an effort is being made to reduce electricity consumption on the ESA IPVC campus?".

The results obtained in the previous question are confirmed by the results obtained with the answers to question "6. Can you give an example of measures that are being implemented at ESA IPVC, to reduce electricity consumption?". In fact, most respondents answered "not being able to give examples of measures being implemented", with the result obtained even surpassing that of the previous question. In other words, most likely, some of the respondents, even if superficially, were aware that something is being done, although they did not know what exactly. Almost all of the remaining respondents, representing around 40% of the answers obtained, presented as examples "the installation of solar panels" and "the replacement of light bulbs with LEDs" as the measures best known by community members.

The answers obtained thus far seem to already indicate the result obtained in the answer to question "7. Do you feel that the set of measures to reduce electricity consumption that is being implemented at ESA IPVC are being communicated in the best way?". As can be seen in the results summarized in Figure 8, a very significant part of the respondents answered that the communication about the measures being implemented was not being done in the best way.

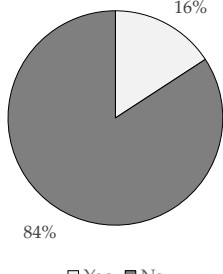
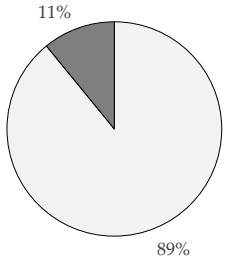

**Figure 8.** Results obtained in the answers to questions "7. Do you feel that the set of measures to reduce electricity consumption that is being implemented at ESA IPVC are being communicated in the best way?" and "8. Do you think it is important that information on energy efficiency measures being implemented at ESA IPVC be communicated to the entire community?".

As can be seen in the results obtained for question 8, almost all respondents answered that it is important that the measures and results obtained are communicated, demonstrating that the lack of knowledge is not due to a lack of interest, but rather due to a lack of information.

Questions 9 and 10 assess, on the one hand, the importance of implementing measures to improve energy efficiency in the ESA IPVC and, on the other hand, the quality of the measures being implemented (Figure 9).

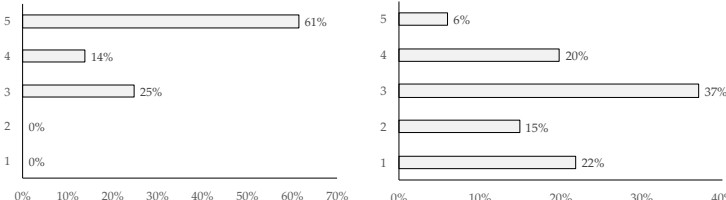

**Figure 9.** Results obtained in the answers to questions "9. On a scale of 1 to 5, how do you rate the need to implement measures to improve energy efficiency?" and "10. On a scale of 1 to 5, how would you rate the quality of the measures that are being implemented to improve energy efficiency?".

As can be seen, a significant majority of respondents considered the need to implement measures to improve energy efficiency on the ESA IPVC campus to be very important to improving the campus's sustainability. On the other hand, a very significant group of respondents classified the quality of the measures that are being implemented to improve energy efficiency on the ESA IPVC campus as being unimportant to moderately important. This situation may be related to the current lack of knowledge about the measures being implemented and about the results obtained by the implementation of these measures.

There was a relative congruence in the answers obtained for question 11, when respondents were asked to present additional measures to improve energy efficiency on the campus. Respondents pointed out as the main measures the "placement of motion sensors in corridors, cloisters, and WCs", "placement of LEDs in the campus' exterior lighting", "control of the use of accessory equipment in the rooms of the residence and cabinets of the central building", and "installation of more solar panels". This list of suggestions demonstrates the community's interest in playing a more active role in changing behavioral habits to reduce energy consumption on campus, as shown in Figure 10.

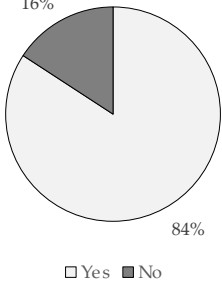

**Figure 10.** Results obtained in the answer to question "12. Are you willing to play an active role in changing behavioral habits to reduce energy consumption at ESA IPVC?".

This willingness to participate more actively in the process is also demonstrated by the results obtained in the answers to question "13. Would you be willing to give up some comfort in order to achieve more ambitious goals to reduce the carbon footprint?", with approximately 2/3 of the members of the academic community answering affirmatively (Figure 11).

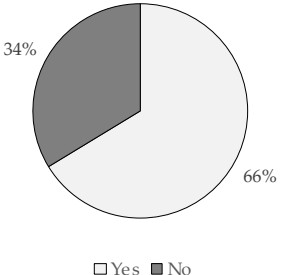

**Figure 11.** Results obtained in the answer to question "13. Would you be willing to give up some comfort in order to achieve more ambitious carbon footprint reduction goals?".

In response to this question, the respondents pointed out the main sacrifices they would be willing to make: "not using elevators, except for people with reduced mobility and for transporting volumes" and "not using heating and other appliances in the rooms of the residence and the cabinets". These suggestions demonstrate the community's willingness to collaborate in the implementation of new measures and to contribute to the successful pursuit of the proposed objectives for the reduction of the campus's carbon footprint. In contrast, in response to the question "15. Would you be available to join an Awareness Brigade for issues related to reducing energy consumption and energy efficiency?" more than 2/3 of respondents answered that they do not intend to actively participate as an awareness-raising agent (Figure 12).

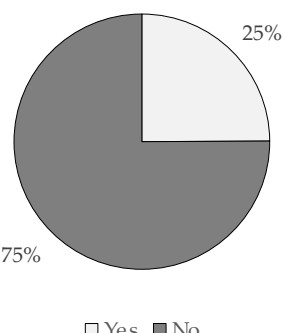

**Figure 12.** Answer to question "15. Would you be available to join an Awareness Brigade for issues related to reducing energy consumption and energy efficiency?".

The last question presented to respondents asked them to choose three out of five measures they would like to see additionally implemented on the ESA IPVC campus. The results obtained are summarized in Figure 13. The respondents chose measure 16.2, which gathered 30% of opinions, followed by measure 16.1, with 23%, and measure 16.3, with 21% of respondents.

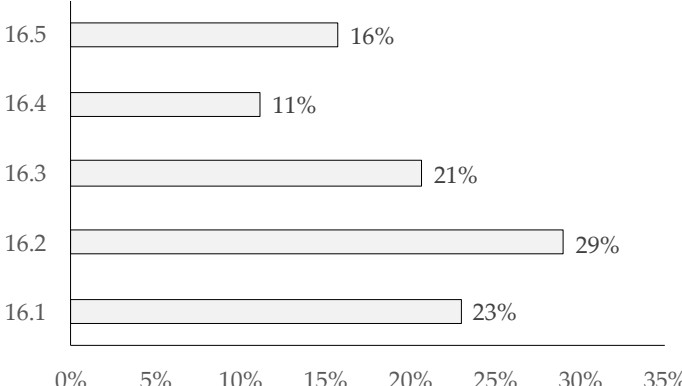

**Figure 13.** Results obtained in response to question "16. From the following measures, select the three that you most agree with being additionally implemented at ESA IPVC: 16.1. Conducting awareness-raising actions for all members of the ESA IPVC community; 16.2. Installation of motion sensors in passage areas and bathrooms, to turn on the lights only when necessary; 16.3. Limit the use of elevators only to people with reduced mobility, or in situations in which volumes are transported; 16.4. Limit the use of non-essential electrical equipment on the ESA IPVC campus, such as radios, televisions, heaters, and fan heaters, among others, in the residence rooms, faculty offices, and other similar places; 16.5. Adjust school hours to make the most of the hours of natural light and avoid the use of artificial light (whenever the situation allows)".

## 5. Discussion

The integration of renewable energy production systems on campuses, namely PV systems, presents itself as a solution that is increasingly attracting more supporters, as it often allows availing of unused spaces, as is the case of buildings' rooftops. These solutions will always be more effective if they not only fulfil their purpose as an advanced model of a renewable energy integration system but also reach environmental research, e-mobility, and energy efficiency [33]. The installation of the PV system on the ESA IPVC campus allows for the generation of a significant amount of energy used in a self-consumption regime. Thus, it responds sustainably to the energy need of the campus as well as the need for carbon emission savings. The PV system installed on the ESA IPVC campus corresponds to 19% of the estimated annual consumption, based on the historical record over the last 11 years. Replacing a significant part of the external electricity supply to the university campus corresponds to a reduction of 21 $tCO_2 \cdot yr^{-1}$ emissions. Despite the environmental analysis demonstrating that the PV system presents encouraging results towards transforming the campus into an energy-efficient and environmentally sustainable community, it is still necessary to proceed with the simultaneous adoption of complementary organizational measures and to raise awareness among users (teaching staff, students, employees, and visitors) in order to change habits. The efficient use of energy significantly contributes to the effective reduction of consumption and the subsequent decrease in carbon dioxide emissions. As Lee [77] points out, this transformation is fundamental for sustainable campus development. At the same time, this transformation contributes to a broader evolution of concepts, values, and attitudes concerning sustainability in individuals, organizations, and general society.

In the case of the ESA IPVC campus, the estimated annual PV production contributes to, as already mentioned, the satisfaction of 19% of the campus's needs. However, this energy production does not occur equitably throughout all the months of the year since it is a form of energy that suffers from a high level of intermittence. Thus, the months corresponding to the highest PV production are in spring and summer, where the days are longer, and energy consumption with lighting is lower. On the other hand, the months in which PV production is lower correspond to Autumn and Winter, when energy needs increase, namely with regards to lighting, since the days are shorter, and energy is used, for example, for portable heating equipment in the professors' cabinets and the rooms of the students' residences.

This perspective shows that energy availability is not directly related to consumption. That is, energy may not be available when it is most needed due to the intermittence of PV energy production. For this reason, real-time knowledge of consumption becomes essential to manage these consumptions correctly. For example, certain activities can be carried out in the periods when PV energy is expected to be produced, in this way the production is consumed and the need to resort to an external supply is avoided. Thus, the combination of technical solutions, such as installing the PV system, which took full advantage of the available space on the rooftop of the university residence, with the adoption of organizational measures would guarantee a more effective and efficient use of available energy. In turn, these measures must be complemented by changing the habits of campus users and supported by raising awareness of the need to evolve towards the sustainable development of the campus and disseminating the results in the form of key performance indicators for the sustainability of the campus, namely, the reduction of energy consumption and the reduction of GHG emissions compared to the pre-installation period of the PV energy production system. This study allows for quantifying the environmental gains obtained from adopting energy self-production measures, as presented in the study carried out at ESA IPVC. However, to guarantee the fulfilment of genuinely sustainable development goals, it is necessary to study other factors, namely those related to the energy efficiency of buildings, and mainly those associated with the change of energy consumption and storage practices. Energy storage systems are becoming increasingly important in the effort to improve the energy efficiency and sustainability of university

campuses [78]. These systems allow for the storage of excess energy generated for later use when energy demand is higher [79]. This helps to mitigate the intermittency of renewable energy sources and ensures a stable and consistent supply of energy [80]. Energy storage systems can also improve energy efficiency by enabling the use of stored energy during peak demand periods, reducing the need to rely on fossil fuel-based power sources [81]. This not only reduces the carbon footprint of the university campus, but also reduces energy costs and increases energy security, providing backup power in the case of power outages, as well as ensuring the continuity of critical operations and services on campus [82]. This is especially important in emergency situations, where the campus must continue to function with minimum disruption [83]. Energy storage systems can also serve as a valuable educational tool, providing students and researchers with hands-on experience in the design, installation, and operation of these systems [84]. This type of practical learning can help to build the workforce of the future and ensure the continued development of sustainable energy solutions [85].

As can be seen from the results obtained in the survey, although there is already a specific recognition of the importance of the measures being implemented on the ESA IPVC campus, there is still an alienation of a very significant section of the members of this community. As can easily be foreseen, the success of the application of non-structural measures of a practical nature to change habits is directly dependent on the members' willingness to collaborate and comply with the proposed measures. This change in practice is based on changing habits, justifying awareness-raising actions in the pursuit of this objective. The results also indicate a considerable lack of knowledge about the ongoing processes, which was also demonstrated by the survey carried out in this study, and which is associated with the lack of communication (or inefficient communication, when existing) of the measures applied or even the results obtained. However, as a form of motivation and to involve more and more members of the academic community in the process, this communication should not, perhaps, be limited to the mere transmission of individual measures or results obtained with the application of any measure. With the willingness shown by community members to play an active role in reducing the carbon footprint, communication must include the presentation of the implementation plan as a whole, with the precise definition of the objectives and with the results presented in the form of performance indicators that community members easily understand, demonstrating the evolution of the entire process.

The implementation of awareness-raising actions and the dissemination of information through digital media, such as social networks, can provide opportunities for receiving feedback from community members. This increased sense of integration and participation can lead to community members becoming active agents of change in their attitudes and behaviors, voluntarily contributing to the success of the measures that have been or will be implemented.

## 6. Conclusions

The sustainability of university campuses, which encompass a vast range of functions and activities that consume significant amounts of energy and water, and generate waste, has become a crucial concern for those responsible for their management. One solution to address this issue is the implementation of sustainable development initiatives, such as the installation of autonomous energy production systems such as photovoltaics (PV). This approach reduces the campus's reliance on external energy sources, often generated from fossil fuels, and reduces greenhouse gas (GHG) emissions. A case study demonstrated that a PV system can meet 19% of the electricity requirements and decrease GHG emissions by 21 $tCO_2 \cdot yr^{-1}$. However, in the analyzed case study, the available space for additional PV installations has been utilized; therefore, improving sustainability indicators requires the adoption of other measures, such as organizational changes and the active engagement of the university community in changing their behavior to improve energy efficiency and optimization.

**Author Contributions:** Conceptualization, L.J.R.N. and A.C.; methodology, L.J.R.N. and A.C.; validation, I.A. and L.J.R.N.; formal analysis, A.C.; investigation, I.A., L.J.R.N. and A.C.; resources, A.C.; data curation, I.A. and L.J.R.N.; writing—original draft preparation, I.A., L.J.R.N. and A.C.; writing—review and editing, I.A., L.J.R.N. and A.C.; visualization, I.A., L.J.R.N. and A.C.; supervision, L.J.R.N. and A.C. All authors have read and agreed to the published version of the manuscript.

**Funding:** This work is a result of the project TECH—Technology, Environment, Creativity and Health, Norte-01-0145-FEDER-000043, supported by Norte Portugal Regional Operational Program (NORTE 2020), under the PORTUGAL 2020 Partnership Agreement, through the European Regional Development Fund (ERDF). L.J.R.N. was supported by proMetheus, Research Unit on Energy, Materials and Environment for Sustainability—UIDP/05975/2020, funded by national funds through FCT—Fundação para a Ciência e Tecnologia. A.C. co-authored this work within the scope of the project proMetheus, Research Unit on Materials, Energy, and Environment for Sustainability, FCT Ref. UID/05975/2020, financed by national funds through the FCT/MCTES.

**Institutional Review Board Statement:** Not applicable.

**Informed Consent Statement:** Not applicable.

**Data Availability Statement:** The data presented in this study are available on per request to the corresponding author.

**Conflicts of Interest:** The authors declare no conflict of interest.

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
