# Peer review of "Preliminary Approach for the Development of Sustainable University Campuses: A Case Study Based on the Mitigation of Greenhouse Gas Emissions"

_sustainability, doi:10.3390/su15065518_

Round 1

Reviewer 1 Report (Previous Reviewer 3)

Preliminary Approach for the Development of Sustainable University Campuses: A Case Study based on the Mitigation of Greenhouse Gas Emissions, I. Araújo, L.J.R. Nunes and A. Curado

Language still needs minor revision. I include an example:  

Line 130

To calculate the power consumption of the ESA IPVC campus was analysed the historical record between December 2010 and January 2020. Based on this record, determined minimum and maximum estimated values for power consumption, taking into consideration the variability of the consumption over the past ten years.

May be: To calculate the power consumption of the ESA IPVC campus we analyzed the historical record between December 2010 and January 2020. Based on this record, we determined minimum and maximum estimated values for power consumption, taking into consideration the variability of the consumption over the past ten years.

General comments:

In my previous review, I suggest to add some storage considerations, answering if the authors considered the possibility of energy storage, actually or in future, and if not, a few comments explaining why they didn´t consider this possibility. Energy storage is an important issue related with energy transition, because it assures that an energy matrix with high renewable sources penetration can work properly. Being this a case study, performed in a University, it would be advisable at least have an opinion about this matter.

Author Response

Comments from Reviewer #1:

Preliminary Approach for the Development of Sustainable University Campuses: A Case Study based on the Mitigation of Greenhouse Gas Emissions, I. Araújo, L.J.R. Nunes and A. Curado

Language still needs minor revision. I include an example: 

Line 130

To calculate the power consumption of the ESA IPVC campus was analysed the historical record between December 2010 and January 2020. Based on this record, determined minimum and maximum estimated values for power consumption, taking into consideration the variability of the consumption over the past ten years.

May be: To calculate the power consumption of the ESA IPVC campus we analyzed the historical record between December 2010 and January 2020. Based on this record, we determined minimum and maximum estimated values for power consumption, taking into consideration the variability of the consumption over the past ten years.

General comments:

In my previous review, I suggest to add some storage considerations, answering if the authors considered the possibility of energy storage, actually or in future, and if not, a few comments explaining why they didn´t consider this possibility. Energy storage is an important issue related with energy transition, because it assures that an energy matrix with high renewable sources penetration can work properly. Being this a case study, performed in a University, it would be advisable at least have an opinion about this matter.

Answer to comments from Reviewer #1: the authors would like to thank once again Reviewer #1 for the time spent reading and commenting on the article submitted to Sustainability. Thank you very much for this additional set of recommendations that the authors tried to address in the best way possible.

Concerning the comments presented, the authors conducted additional proofreading of the entire manuscript to improve the English language level, as suggested by Reviewer #1. Concerning the general comments, and more precisely the subject of energy storage, in the initial project it was not intended because it would overpass the allowed budget. All the investment was done by the Ministry of Educational Affairs and was more or less equal to the majority of the institutions. A first approach was intended to fit all the campuses with similar equipment and in a future tendering opportunity, each university will be able to apply for further needs following the studies and assessments being conducted right now. For this reason, energy storage was not discussed in the study. However, to address the recommendation presented by Reviewer #1, the authors now mentioned the hypothesis in the discussion section for a future upgrade of the installation.

Thank you very much once again. The authors expect that now the article can be accepted for publication with the support of Reviewer #1.

Reviewer 2 Report (Previous Reviewer 2)

This is a well structured and well presented paper, but I find it difficult to understand the novelty presented.

From a technical point of view (I am an engineer), the work on the prediction of PV needs better justification and explanation - I do not understand why you are doing it, and why you have chosen your particular source.

From a social science point of view (not my expertise), the results look obvious ie they could have been predicted before the survey was undertaken. For me, a more interesting approach would be to quantify in monetary terms what the savings are, and then see if the respondents had more engagement with the results.

In conclusion, whilst I understand that a lot of effort has gone into preparing the paper, I still do not understand the novel findings from the work that make it worthy of publication.

Author Response

Comments from Reviewer #2:

This is a well structured and well presented paper, but I find it difficult to understand the novelty presented.

From a technical point of view (I am an engineer), the work on the prediction of PV needs better justification and explanation - I do not understand why you are doing it, and why you have chosen your particular source.

From a social science point of view (not my expertise), the results look obvious ie they could have been predicted before the survey was undertaken. For me, a more interesting approach would be to quantify in monetary terms what the savings are, and then see if the respondents had more engagement with the results.

In conclusion, whilst I understand that a lot of effort has gone into preparing the paper, I still do not understand the novel findings from the work that make it worthy of publication.

Answer to comments from Reviewer #2: The authors would like to thank Reviewer #2 for the additional time spent reading and commenting on the article. Thank you as well for the opportunity to clarify the additional points raised because this clarification will avoid misunderstandings for the common reader in the future.

Thank you very much for considering that this is a well-structured and well-presented paper, and now the authors will try to explain some of the questions raised by Reviewer #2.

Concerning the novelty, in fact, it is an article that approaches the sustainability assessment in a university campus where some renewable energy production systems have been installed. Here the authors entirely agree with Reviewer #2, and the novelty is not much. However, as happens with many articles published year after year, sometimes the novelty is not properly in the achievements, but in the way how the conclusions can be achieved. For instance, Reviewer #2 considers that the results could be concluded without the survey. However, in science, it is not possible (or at least it is not recommended) to predict conclusions or results without having support such as the one provided by the survey. Now, and this is a novelty, are presented results about the considerations and opinions that the users of a campus have, and this can, with a large probability, be expanded to other similar situations. The novelty of the present work can be pointed as being the combination of the usual practices to improve the sustainability indexes, such as the installation of a renewable energy system in a campus, together with the assessment of public opinion, which will be taken into consideration in future improvements and interventions.

Concerning the methodology used in this work, the authors understand the opinion of reviewer #2. However, the authors also have the opinion that once the prediction of the energy consumption is not linear on a yearlong basis, and for that reason intended that should be more correct to use a stochastic approach instead of a statistical one. With the monthly variations the error introduced in the calculations would be much higher if a mere statistical approach was used. The authors expect that Reviewer #2 can accept this explanation.

Concerning the economic analysis recommended by Reviewer #2, it is a subject that would be very difficult to apply in the current situation. Being a public university, with the installation of the PV system financed by public funds, the campus cannot sell the extra production to the energy grid. This situation is due to avoid concurrent problems with the private sector. For example, to receive energy at a lower price, the energy companies could foment the installation of renewable energy systems in campuses and other public buildings to inject more energy into the grid forcing the private sector to lower the energy prices. Also, energy acquisition is made in a different way for being a public institution, and for that reason, it is difficult to compare.

The authors expect that now Reviewer #2 can recommend the article in Sustainability in its current form. Thank you very much.

Reviewer 3 Report (New Reviewer)

The paper is all about university campuses that are seen as high-energy-consuming ecosystems. Their amount of waste (of the different resources) are as big as the amount of waste compared to a small town. So it is about renewable energy and other kind of renewable resources, so the campus life of the students could be way more sustainable.

Overall the Paper is well written. It is really detailed and has enough information so every reader can understand it even though they are not familiar with the topic. It is quite informative and explains everything really well without being too much information at once.

The overview that is given in the whole paper is really understandable. The conclusion of the paper is also well written but could be longer so every result could be mentioned. Also there should be an outlook on future research.

 Major comments:

-          Literature Review

ð  Is not a chapter on its own (inside introduction)

ð  Add a short, half to one page section with the title "A brief literature review"

ð  Good length

-          Really good that the different calculations and methods are described in the beginning of the paper

-          Interview questions:

ð  Are described and mentioned in the paper

ð  Gives an overview over the questions

ð  Reader can see how the results came about

-          Results:

ð  Are very detailed

ð  Well written

ð  Easy to understand

Minor comments:

-          Literature review and the Introduction should be two separate chapters. Refer to https://doi.org/10.3390/su12093880 and https://doi.org/10.1016/j.seps.2022.101347 for the enrichment of a general discussion.  The first paper elaborates on the role of environmental regulation stringency and the second one on the role of civic engagement.

-                   Conclusion is als well written

ð  Should be more extensive

ð   Also an outlook an new studies and research is not given

Author Response

Comments from Reviewer #3:

The paper is all about university campuses that are seen as high-energy-consuming ecosystems. Their amount of waste (of the different resources) are as big as the amount of waste compared to a small town. So it is about renewable energy and other kind of renewable resources, so the campus life of the students could be way more sustainable.

Answer to comments from Reviewer #3: the authors would like to thank Reviewer #3 for the additional comments and recommendations. Thank you very much for the constructive considerations that much helped the authors to improve the final quality of the article. The authors expect that now the article can be accepted in its current form.

Overall the Paper is well written. It is really detailed and has enough information so every reader can understand it even though they are not familiar with the topic. It is quite informative and explains everything really well without being too much information at once.

Answer to comments from Reviewer #3: Thank you very much for considering that the paper is well written, really detailed and that has enough information so every reader can understand it even though they are not familiar with the topic. Thank you for considering that iti is quite informative and explains everything really well without being too much information at once.

The overview that is given in the whole paper is really understandable. The conclusion of the paper is also well written but could be longer so every result could be mentioned. Also, there should be an outlook on future research.

Answer to comments from Reviewer #3: Thank you very much for considering that the overview that is given in the whole paper is understandable and that the conclusion of the paper is also well written. Considering the recommendation presented by Reviewer #3, the authors extended the conclusions and included now an outlook on future research.

 Major comments:

-          Literature Review

ð  Is not a chapter on its own (inside introduction)

ð  Add a short, half-to-one-page section with the title "A brief literature review"

ð  Good length

Answer to comments from Reviewer #3: the authors entirely agree with Reviewer #3 and included now a section titled “A Brief Literature Review” as recommended by Reviewer #3.

-       Really good that the different calculations and methods are described in the beginning of the paper

Answer to comments from Reviewer #3: Thank you very much for this comment. The authors agree with the opinion of Reviewer #3. However, analyzing the complexity of some data and calculation methods, the authors have the opinion that explaining the origin of the data and the methodology used as the section emerges can facilitate the readers to understand the situation without the need to return in the text to see how it was done. The authors kindly ask Reviewer #3 to accept this explanation.

-          Interview questions:

ð  Are described and mentioned in the paper

ð  Gives an overview over the questions

ð  Reader can see how the results came about

Answer to comments from Reviewer #3: Thank you very much for this set of positive comments.

-          Results:

ð  Are very detailed

ð  Well written

ð  Easy to understand

 Answer to comments from Reviewer #3: Thank you very much for this set of positive comments.

Minor comments:

-      Literature review and the Introduction should be two separate chapters. Refer to https://doi.org/10.3390/su12093880 and https://doi.org/10.1016/j.seps.2022.101347 for the enrichment of a general discussion.  The first paper elaborates on the role of environmental regulation stringency and the second one on the role of civic engagement.

Answer to comments from Reviewer #3: Thank you very much for this recommendation. The authors accepted the suggestion and included both references in the text. 

-                   Conclusion is also well written

ð  Should be more extensive

ð   Also an outlook an new studies and research is not given

Answer to comments from Reviewer #3: Thank you very much for this set of positive comments. The authors expect to have addressed properly the question raised by Reviewer #3 and expect that now the article can be accepted in its current form. Thank you very much.

Round 2

Reviewer 3 Report (New Reviewer)

The paper is perfect now! Congratulations!

This manuscript is a resubmission of an earlier submission. The following is a list of the peer review reports and author responses from that submission.

Round 1

Reviewer 1 Report

Thank you for the opportunity to review this paper. 

After careful consideration, I offer the following advice.

1. The English needs a lot of work, there is a lot of use of strange sentences and word choice.

2. I think that the consideration only of solar because it is already on the roof, and because the building is heritage listed is incredibly short sighted - surely a comparison to locally located wind power, biomass or any other source of RE would be valid here?

3. The presentation of the survey results is haphazard, the graphs are very hard to read - use a simple contrasting color choice here.

3a. Also, with the survey results, although you outline the survey instrument, you only describe the nuanced responses in your words - these should also be visualized - it feels like you are trying to save time or effort here.

4. The conclusions are lackluster - what is the take home message for us?

Reviewer 2 Report

This paper describes the authors' findings regarding the installation of a PV system on a university campus. The paper presents results concerning the prediction of consumption and the prediction of generated PV energy, and results from a questionnaire given to building users. The main objective of the work was to “study how the installation of a PV system impacts the campus's energy consumption as a case study”.

I struggle to find the novelty presented in the paper. The consumption and energy forecasting techniques are not described in any great detail, and I do not believe them to be original. The results presented do not seem to demonstrate the effectiveness of the techniques employed over other techniques presented in the wider literature, so I am not clear why they are presented in this paper. Measurements before and after the installation of PV would be sufficient to show the impact of the PV system – why is the work on prediction required?

The questionnaire regarding the impact of the awareness of sustainable development is potentially interesting as a subject for the paper, but unfortunately the presentation and interpretation of the results is very superficial. If the building’s users had been involved in the design and implementation of sustainability measures from the outset, then perhaps a more meaningful engagement may have resulted. However the paper (to me) demonstrates that end user engagement is essential as part of planning new building operation measures for sustainability, and indeed more can be achieved by engagement rather than investment in technology, if innovative user engagement measures are employed from the outset. Unfortunately, this opportunity has been missed in this instance which is disappointing.

Reviewer 3 Report

Preliminary Approach for the Development of Sustainable University Campuses: A Case Study based on the Mitigation of Greenhouse Gas Emissions, I. Araújo, L.J.R. Nunes and A. Curado

This is a very interesting work, well-discussed, well-founded and well-written. I suggest to add some storage considerations. Did the authors consider the possibility of energy storage, actually or in future? If not, why?  

Language needs minor revision; some observations are included here as examples.

Line 28

If on the one hand, university campuses can contribute significantly to the development of societies through this function of creating and disseminating knowledge [5]. On the other hand, due to the enormous scale that campuses can reach, they can become centers of high-level of energy use, high-level of waste production, high-level of water consumption, as well as high-level of maintenance requirements for buildings, infrastructures, and green areas [6–8].

If on the one hand, university campuses can contribute significantly to the development of societies through this function of creating and disseminating knowledge [5], on the other hand, due to the enormous scale that campuses can reach, they can become centers of high-level of energy use, high-level of waste production, high-level of water consumption, as well as high-level of maintenance requirements for buildings, infrastructures, and green areas [6–8].

Line 121

A set of restrictions regarding interventions in the building itself, but also its surroundings.

A set of restrictions regarding interventions in the building itself, but also in its surroundings.

Line 122

The ESA IPVC campus occupies a significant area, with approximately 17 hectares, since in addition to the various buildings, it also includes the area of the experimental farm, where different crops are produced, such as vines, olive groves, and other crops in rotation, but also where there are areas dedicated to the animal production of indigenous breeds of cattle.

The ESA IPVC campus occupies a significant area, with approximately 17 hectares, since in addition to the various buildings, it also includes the area of the experimental farm, where different crops are produced, such as vines, olive groves, and other crops in rotation, but also where there are areas dedicated to the animal production of indigenous breeds of cattle.

Line 134

Based on this record, was created an estimated value.

Based on this record, it was created an estimated